# Information Theoretic Approaches for Testing Missingness in Predictive Models

**Shreyas Bhave** [1]   **Rajesh Ranganath** [2]   **Adler Perotte** [1]

## Abstract

In predictive modeling, missing data can often result in learning biased models despite application of imputation approaches. Therefore, it is important to assess the missingness process of the data. We present hypothesis tests for assessing these dependencies: MI-MCAR (mutual information for missing completely at random) and MI-US (mutual information for unobserved sources). MI-MCAR tests marginal independence between the missingness pattern and the the data matrix, while MI-US is a conditional randomization test (CRT) to test the dependence of the missingness pattern on unobserved sources. These methods can be applied to heterogeneous data types and can serve to identify missingness pathologies in data which specifically affect performance for regression tasks. We evaluate our methods on simulated and pseudo-simulated datasets and show that we are able to identify data which suffers from missingness due to unobserved sources.

## 1. Introduction

Missing data are ubiquitous in many research areas and are often overlooked when building predictive models. In many cases, practitioners choose to completely remove samples with missing data elements or may use one of the many imputation methods such as multiple imputation. Despite the application of imputation approaches, models can be biased especially in the case of data that is missing not at random (MNAR). Critically assessing the dependencies in the underlying missingness process can lead to a deeper understanding of observational data as well as inform modeling approaches and imputation strategies. Past work has focused on modeling the underlying missingness process by leveraging graphical models (Mohan et al., 2013; Sh-

pitser et al., 2015). There has also been work on developing statistical tests to test for certain categories of missingness. Little's test (Little, 1988; Li, 2013) uses a chi-square statistic with a null hypothesis that tests a necessary condition for missing completely at random (MCAR) data. Since Little's test, many other tests have been proposed to test the MCAR assumption including likelihood ratio tests with a missing at random (MAR) alternative (Lipsitz et al., 1994) as well as non-parametric extensions (Chen & Little, 1999).

In this work, we present information theoretic approaches to testing for missingness in the context of predictive models. We highlight mutual information and conditional mutual information as favorable measures for testing independence. Mutual information approaches are invariant to smooth, invertible transformations and thus are nonparametric in a specific sense (Weihs et al., 2018). Additionally, mutual information can be decomposed into entropy and conditional entropy terms which makes modern density estimation approaches amenable for its estimation. We also note that neural density estimation approaches (Bishop, 1994; Germain et al., 2015) can accommodate quantitative and categorical data and as such a test statistic based on mutual information could serve as a more general test for independence.

We summarize our contributions as: (1) We propose *MI-MCAR* (mutual information for missinng completely at random), which tests for an MCAR assumption by leveraging density estimation approaches to estimate mutual information between the data matrix ($\mathbf{X}$) and the missingness pattern ($\mathbf{R}$) (2) We propose *MI-US* (mutual information for unobserved sources) which identifies dependence of the missingness pattern on unobserved sources by testing for dependence between the missingness pattern ($\mathbf{R}$) and the response ($\mathbf{Y}$) given the observed data ($\mathbf{X}$) in the specific setting of a predictive model where the response is fully observed.

The methods in this work could help practitioners gain additional insights into whether their data violates certain assumptions that are normally taken for granted (MCAR or MAR) specifically in the context of predictive modeling and handle them accordingly. This could entail searching for external features which may explain this dependence to add them to the feature set, changing data collection practices where possible or removing features.

---

[1]Department of Biomedical Informatics, Columbia University, New York, New York, USA [2]The Courant Institute, New York University, New York, New York, USA. Correspondence to: Shreyas Bhave <sab2323@cumc.columbia.edu>.

*Presented at the first Workshop on the Art of Learning with Missing Values (Artemiss) hosted by the $37^{th}$ International Conference on Machine Learning (ICML).* Copyright 2020 by the author(s).

## 2. Background

### 2.1. Mutual Information and Conditional Mutual Information

Mutual information is a natural measure of dependence between two variables. It is equivalent to the KL divergence between the joint and product of the marginal distributions of two random variables. Furthermore, it can be split into a marginal and conditional entropy which is useful when doing density estimation.

$$I(X;Y) = \int_Y \int_X P_{X,Y}(x,y) log \frac{P_{X,Y}(x,y)}{P_X(x)P_Y(y)} \quad (1)$$
$$= H(Y) - H(Y|X)$$

The definition of conditional mutual information is very similar.

$$I(X;Y|Z) = \int_X \int_Y P_{X,Y|Z}(x,y|z) log \frac{P_{X,Y|Z}(x,y|z)}{P_{X|Z}(x|z)P_{Y|Z}(y|z)}$$
$$= H(Y|Z) - H(Y|X,Z)$$
$$(2)$$

### 2.2. Tests for MCAR

Rubin originally introduced the different categories of missing data by summarizing them as a set of independence assumptions (Rubin, 1976; Roderick et al., 2002). Let $X$ denote the data matrix where $X_{obs}$ is the observed portion and $X_{mis}$ is the missing portion of data. Let $R$ be the matrix of missingness indicators. Given this, the assumptions for MCAR and MAR can be formulated as follows as in Rhoads (2012):

$$P(X_{mis}|X_{obs}, R)P(X_{obs}|R) = P(X_{obs}, X_{mis}) \quad (3)$$

$$P(X_{mis}|X_{obs}, R) = P(X_{mis}|X_{obs}) \quad (4)$$

From this particular formulation, it is clear from (3) that a MCAR assumption requires both that the observed data is marginally independent of the missingness pattern *and* that the missing data is conditionally independent of the missingness pattern given the observed data. On the other hand, as in (4) a MAR assumption only requires the latter. Therefore, all MCAR data is MAR while the converse is not true.

*MI-MCAR* is fundamentally testing the marginal independence assumption of MCAR and thus can serve to confirm that data is not MCAR. We contend that the true advantage of this test is that it leverages mutual information as the test statistic which is invariant to transformations of the data and can be useful with combinations of different data types with modern density estimation approaches.

While it is not possible to distinguish between a MAR and MNAR assumption from just observed data, we propose a method *MI-US* to test for the case of dependence between the missingness pattern and *unobserved sources* which have a significant effect on the response variable ($Y$) in the context of a regression problem. We define *unobserved sources* to mean any feature which is missing in the data $X_{mis}$ or missing as a result of not being part of the feature set. We note that definitions of MNAR often only consider missingness inside of the feature set, or $X_{mis}$.

## 3. MI-MCAR: Test for MCAR

Let $X \in R^{NxP}$ denote the data matrix with some set of missing values where $R \in \{0,1\}^{NxP}$ denotes the missingness pattern matrix, and $r_i$ and $x_i$ refer to specific points.

Let $X_{imp}$ denote an imputed data matrix where missing values are imputed using a multiple imputation approach. We estimate the mutual information between $R$ and $X$ by estimating the entropy and conditional entropy terms as follows:

$$\hat{I}(R, X_{obs}) = \hat{H}(R) - \hat{H}(R|X_{obs})$$
$$\hat{H}(R) = -\frac{1}{N} \sum_{i=1}^{N} log(p_R(r_i))$$
$$\hat{H}(R|X_{obs}) = -\frac{1}{N} \sum_{i=1}^{N} log(p_{R|X_{imp}}(r_i|x_i)) \quad (5)$$

We note that if $X_{imp}$ is constructed via a multiple imputation approach, then the estimated conditional entropy will be $\hat{H}(R|X_{obs})$. Using multiple imputation preserves this relationship: $p(R|X_{obs}, X_{imp}) = p(R|X_{obs})$.

Recent work has highlighted the advantages of using mutual information to develop a test for independence (Berrett & Samworth, 2019). We use a similar approach to construct this test, while noting that we estimate a marginal and conditional entropy term and use different approaches for density estimation.

The null hypothesis in this setting is that the joint distribution is not significantly different from the product of the marginals. In order to obtain a p-value for this test, we can construct a dataset $\{\{\tilde{R}_i^b\}_{i=1}^N\}_{b=1}^B$ which consists of $B$ separate samples of the marginal distribution, $p_R$. We can use this to construct a pseudo-dataset $\{(X_{obs}, R^b)\}_{b=1}^B$ and estimate mutual information for each of these $\hat{I}^b(R^b, \tilde{X}_{obs})$.

In order to obtain a p-value for this test we can then use the following:

$$\hat{ct} = \sum_{b=1}^{B} \mathbb{1}\left(\hat{I}(R, X_{obs}) \leq \hat{I}^b\right)$$
$$\hat{p} = \frac{1}{B+1}\left(1 + \hat{ct}\right) \quad (6)$$

This is a p-value for a one-sided test for the independence condition between $R$ and $X_{obs}$ with the alternative being that they are not independent. We note that the distribution of the count $\hat{ct}$ under the null hypothesis is discrete uniform with support from $0$ to $B$. This results in the addition of a 1 in the numerator and the denominator and produces a test with the correct significance level, preventing the possibility of a p-value of 0 (Phipson & Smyth, 2010).

While it is necessary to refit the conditional density estimate for each null sample, we note that it is not necessary to obtain a new density estimate for $R$, nor is it necessary to estimate a density over the observed data $X_{obs}$.

Here, we have presented a test for MCAR which is flexible for many data types, robust to transformations and relatively simple to estimate a p-value for.

## 4. MI-US: Test for Dependence on Unobserved Sources

While it is not possible in general to distinguish between MAR and MNAR, in the case of a regression problem with a fully observed outcome $Y$ we can leverage the information about the missing data in $Y$ as a surrogate for $X_{mis}$.

Instead of testing the conditional independence between $X_{mis}$ and $R$ given $X_{obs}$, we can instead test the conditional independence between $Y$ and $R$ given $X_{obs}$ as in (7).

$$P(R, Y | X_{obs}) = P(R | X_{obs}) P(Y | X_{obs}) \quad (7)$$

If we can reject the null hypothesis, this indicates that there is residual information between the missingness pattern $R$ and the response $Y$ which is unexplained by $X_{obs}$. This can be attributed to *unobserved sources* including but not limited to $X_{mis}$ as shown in Figure 1, with other unobserved sources $U$ also possibly influencing $Y$ and $R$. While this test will not be able to capture all unobserved sources that affect the missingness process, we argue that it captures all the sources that are relevant for the purposes of prediction by measuring information against $Y$.

The specific formulation of the test is a conditional randomization test (CRT) (Candes et al., 2018) using conditional mutual information as a test statistic. Previous work has outlined a criterion for why conditional mutual information is a *proper* test statistic in this setting (Sudarshan et al., 2020). See Appendix A for details.

In our context, we can use the same setup as in Candes et. al. but instead consider the conditional independence between $Y$ and $R$ and use conditional mutual information as the test statistic. The null samples in our setting $\tilde{R}$ are drawn from the distribution $p(R | X_{obs})$.

$$I(R, Y | X_{obs}) = H(Y | X_{obs}) - H(Y | X_{obs}, R)$$
$$I_{null}(\tilde{R}, Y | X_{obs}) = H(Y | X_{obs}) - H(Y | X_{obs}, \tilde{R}) \quad (8)$$

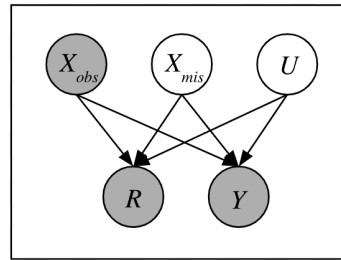

*Figure 1.* Data generation process of the outcome $Y$ and the missingness pattern $R$. If there is MNAR missingness or missingness due to other unobserved sources $U$ which has a significant effect on the outcome, the test will capture this by testing the conditional independence of $Y$ and $R$ given $X_{obs}$.

Given this decomposition of the conditional mutual information, we can see that the first entropy term cancels and we are left with the following p-value:

$$\hat{ct} = \sum_{b=1}^{B} \mathbb{1} \left( \hat{H}(Y | X_{obs}, R) \geq \hat{H}(Y | X_{obs}, \tilde{R}^b) \right)$$
$$\tilde{R}^b \sim P(R | X_{obs}) \quad (9)$$
$$\hat{p}_{unob} = \frac{1}{B+1}(1 + \hat{ct})$$

In order to obtain a p-value for this test, we must resample $\tilde{R}$ and then build a new model for each conditional distribution $P(Y | X_{obs}, \tilde{R}^b)$.

Assuming we have a model specification for each of the terms above including $P(Y | X_{obs}, R)$, $P(R | X_{obs})$ and $P(Y | X_{obs}, \tilde{R})$, we can estimate the p-value as $\hat{p}_{unob}$.

We can see that this p-value is being calculated by resampling $R^b$ from a conditional distribution and counting the number of times the entropy of the conditional model of $Y$ is greater when using the true missingness pattern $R$. This is directly testing the performance gain from including the missingness pattern $R$ in a model for $Y$.

## 5. Experiments

### 5.1. MI-MCAR Simulated Data

We test the MI-MCAR method on a combination of binary data and continuous normal data with different numbers of features. In order to simulate each kind of missingness, we specify a set of variables which are always present and a set that can be missing. Subsequently, we randomly initialize a linear model to model the functional dependence between the data and the missingness pattern. Depending upon the type of missingness, we expose the model to just fully observed data (MAR) or partially observed data (MNAR). See Appendix B for further details.

Table 1. MI-MCAR Empirical rejection rate with different numbers of features on heterogeneous data (binary and continuous)

| $f$ | MCAR | MAR | MNAR |
|-----|------|-----|------|
| 10 | 0.02 | 1.00 | 0.98 |
| 50 | 0.04 | 1.00 | 1.00 |
| 100 | 0.02 | 1.00 | 1.00 |

Table 2. MI-US empirical rejection rate under different missingness simulations with different number of features

| $f$ | MCAR | MAR | MNAR |
|-----|------|-----|------|
| 10 | 0.02 | 0.06 | 0.87 |
| 50 | 0.05 | 0.03 | 0.96 |
| 100 | 0.03 | 0.02 | 0.94 |

We run 100 different seeds of each simulation and run the MI-MCAR test to obtain a p-value for each seed. We present the results in Table 1 as empirical rejection rates at significance level $\alpha = 0.05$. See Appendix D for p-value distributions.

### 5.2. MI-US Simulated Data

For the simulation from unobserved sources, we simulate a logistic model where the output is a binary variable $Y$. The output is regressed on a set of continuous features. We simulate missingness in the same way as in the prior experiment. We use simple logistic regression models for $P(Y|X_{imp}, R)$ and $P(Y|X_{imp}, \tilde{R})$. In order to estimate $P(R|X_{imp})$ we use a logistic model. See Appendix B for further details.

We run 100 different seeds of each simulation and run the MI-US test to obtain a p-value for each seed. We present the results in Table 2 as empirical rejection rates at significance level $\alpha = 0.05$. See Appendix D for p-value distributions.

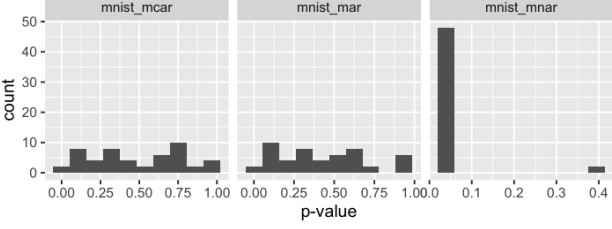

Figure 2. MNIST semi-synthetic dataset p-value distributions under different missingness assumptions across 50 seeds. The distributions on MCAR and MAR data are near-uniform while MNIST datasets consistently reject the null.

### 5.3. MNIST Semi-synthetic Data

In order to evaluate the proposed tests on real data, we use MNIST data with the missingness patterns simulated. The prediction task is the classic digit classification task. We select a patch over which we simulate a missing pattern for every digit based on values within the patch (MNAR) or outside the patch (MAR) just as in prior simulations. We impute the missing data using MissForest (Stekhoven & Bühlmann, 2012) and then run the MI-US test. If the missing patch of pixels is predictive of the output, we can determine that in cases where missingness is MNAR there is dependence on unobserved sources (missing data) while in cases where there is MAR this dependence does not exist. See Appendix C for further details.

We run 50 different seeds for each missingness type, presenting the distribution of the p-values in Figure 2.

## 6. Discussion and Conclusion

The results on simulated data show that MI-MCAR rejects MCAR data near the expected rate of rejection, while it is does not reject with MAR and MNAR simulated data. These results provide evidence that this test can be effective with heterogeneous data and could serve as an alternative to Little's test in cases where data types are mixed.

The simulations for MI-US in Table 2 shows that the test is able to reject in cases of MCAR and MAR near the expected rate of rejection, while MNAR simulations show some possibility of type II error. In general, this can be attributed to a number of causes including: weak influence of missing data on $Y$, missingness model has weak dependency structure on missing data, missing data can still be effectively estimated from observed data. In each of these cases, it is not necessarily unfavorable to be unable to reject the null as practitioners need not be as concerned from a predictive modeling perspective.

The MNIST semi-simulated results shown in Figure 2 demonstrate the promise of these methods on real data. We run 50 different seeds representing different missingness pattern simulations. It is clear from Figure 2 that the p-value distribution in the case of MCAR and MAR data is near uniform while the test rejects in all but one case on MNAR simulated missingness.

The main limitation of running these tests is that they are computationally expensive as they require retraining a model across multiple resamples. Given that model training isn't prohibitively expensive, running MI-US could reveal important insights about missingness pathologies that have a significant effect on performance. In future work, we hope to scale the results to higher dimensions with more seeds as well as apply the test on real-world clinical data.

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

# A. Conditional Randomization Tests

A CRT is defined as follows as in Candes et al. in the setting of feature importance.

Assume we have a set of independent samples where $X$ is a vector and $Y$ is the outcome $\{X_i, Y_i\}_{1 \leq i \leq n}$ assembled as a data matrix $\mathbf{X}$ and response vector $Y$. Further, a statistic $T_j(\mathbf{X}, y)$ is defined to test whether $X_j$ and $Y$ are conditionally independent given $X_{-j}$ (all variables except $j$). With this setup, a new matrix $X^b$ is simulated by sampling from the conditional distribution: $p(X_j | X_{-j})$. With this setup, the one-sided p-value is as follows:

$$P_j = \frac{1}{B+1} \left( 1 + \sum_{b=1}^{B} \mathbb{1} \left( T_j(X^b, y) \geq T_j(X, y) \right) \right) \tag{10}$$

In our context, we can use the same setup as in Candes et. al. but instead consider the conditional independence between $Y$ and $R$ and use conditional mutual information as the test statistic. The null samples in our setting $\tilde{R}$ are drawn from the distribution $p(R | X_{obs})$.

$$I(R, Y | X_{obs}) = H(Y | X_{obs}) - H(Y | X_{obs}, R)$$
$$I_{null}(\tilde{R}, Y | X_{obs}) = H(Y | X_{obs}) - H(Y | X_{obs}, \tilde{R}) \tag{11}$$

We notice that when using the same construction for the p-value as above, the first entropy term cancels. This results in the p-value in (9).

# B. Simulation Specification

## B.1. MI-MCAR Simulation

For the MI-MCAR simulated datasets for each seed, we simulate a set number of variables $f = \{10, 50, 100\}$ with dataset size $N = 1000$, wherein each simulation, half of the variables are iid Bernoulli with a randomly specified parameters and the other half are drawn from a multivariate

normal distribution with a random mean and covariance matrix.

We use a logistic model to estimate $p(R|X_{obs})$, however we also ran tests with a masked autoencoder approach (MADE), which can be used in a more general setting where the missingness process is unknown. For the purposes of this test, we use $B = 50$.

### B.1.1. MISSING DATA

For each dataset, we randomly specify half of the features to be fully observed and the other half to be possibly missing. Then, we aim to simulate a missingness pattern over the partially observed set of features.

In order to simulate MCAR missingness, we simply draw from a Bernoulli distribution with missingness probability 0.5.

For simulating MAR and MNAR missingness patterns, we randomly initialize a linear model which generates a missingness matrix of probabilities. For MAR missingness, we can simply input the fully observed set of features into a randomly initialized linear model with a sigmoid activation to generate the missingness probabilities. Subsequently, we sample a Bernoulli distribution with these probabilities to generate the missingness pattern. For MNAR missingness, we input the partially observed features and output a missingness pattern thus creating dependence on missing data.

### B.2. MI-US Simulation

For the MI-US simulated datasets for each seed, we simulate the output as a binary variable $Y$. The output is regressed on a set of continuous features which are sampled from a multivariate normal distribution with a random mean and covariance matrix. The conditional model specification of $Y$ is a simple logistic regression. We use a simple logistic model to estimate $p(R|X_{obs})$ but also run some tests with mixture density networks. These networks may be used when the missingness process is more complex.

The missingness patterns are simulated in the same way as with the MI-MCAR simulations. After fitting (and refitting) each of these models, we can calculate the entropy directly as the negative log likelihood loss and count the number of resampled datasets which have smaller entropy than the model with the true missingness pattern to obtain a p-value. For the purposes of this test, we use $B = 50$.

## C. MNIST Semi-Synthetic Specification

We use a subset of the MNIST dataset with $N = 10000$. In order to simulate missingness in the context of image data, we specify a 14 by 14 mask at the center of the image to

simulate a missingness pattern over. The outer edges of the image are considered fully observed.

In order to simulate the missingness pattern for MCAR, we simply generate a binary mask with a missingness probability of 0.5.

For generating MAR missingness, we use a similar approach as previous experiments with a randomly initialized linear model which takes as input the fully observed (flattened) data and outputs Bernoulli parameters over the missingness mask. We can then threshold the missingness probabilities in order to generate a missingness pattern.

In order to generate MNAR missingness, we use a simpler approach than past approaches which is to identify all pixels in the missing 14 x 14 region which are above a certain threshold in pixel value (0.2 in our case) and then randomly sample from a Bernoulli with a fixed probability (0.9 in our case) for each of these points. All pixels below the threshold can be randomly sampled with a lower fixed probability (0.1 in our case). This creates MNAR dependence which directly depends on the missing values themselves.

We specify a basic CNN model with two convolutional layers, relu activations with max pooling and dropout layers in between as the model for $P(Y|X_{obs}, R)$. The input is multi-channeled with the $X_{imp}$ and $R$ as the two channels.

For $P(R|X_{obs})$ we use a simple logistic model but also experiment with mixture density networks which can be used in more general cases where the missingness process is not known.

For these experiments, we use $B = 30$ as it is more expensive to train many CNN models.

## D. P-value Distributions

In this section, we include the p-value distributions for each of our simulations to show that the null distributions closely reflect a uniform distribution.

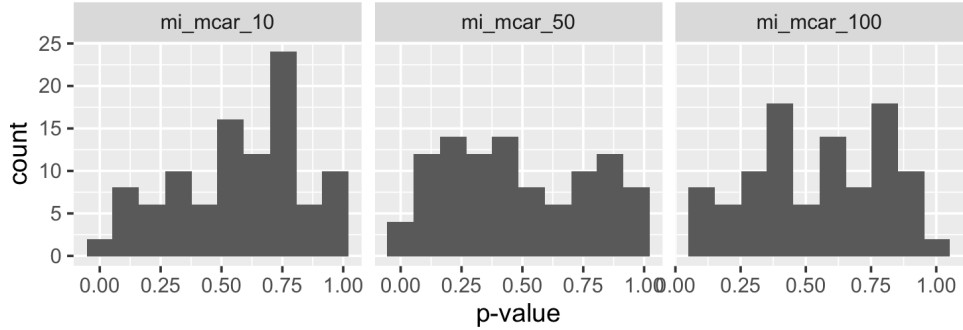

*Figure 3.* MI-MCAR simulation p-value distributions for MCAR data with different feature numbers.

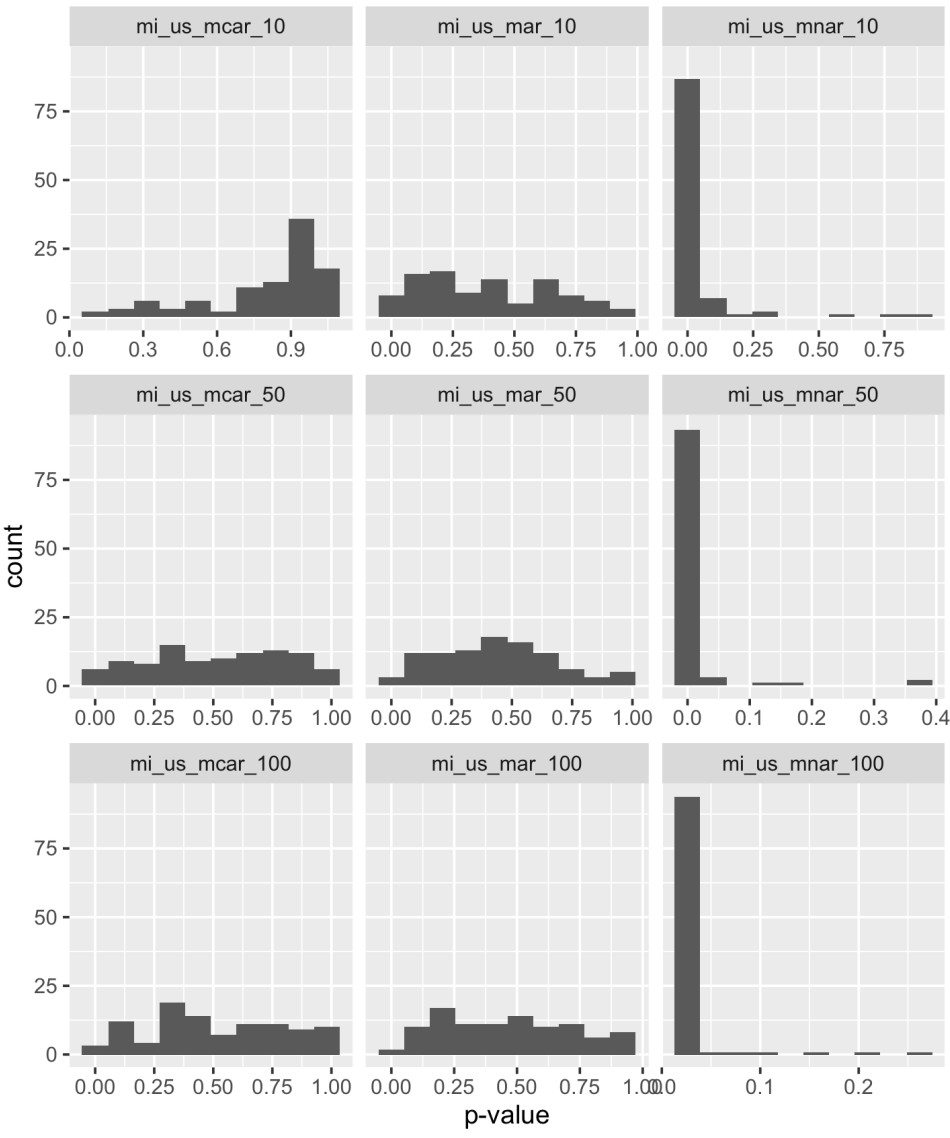

*Figure 4.* MI-US simulation p-value distributions on different feature sizes under different missingness pattern assumptions. The distributions are near uniform for MCAR and MAR simulated data while MNAR rejects the null hypothesis a high percentage of the time.