# OpenReview forum: "Information Theoretic Approaches for Testing Missingness in Predictive Models"
_ICML.cc/2020/Workshop/Artemiss — ICML Artemiss 2020_

### Official Review · AnonReviewer2 · 2020-06-22
**The authors present hypotesis tests for testing the missingness process of the data (MCAR, MAR or MNAR) in predictive models.**

**Rating:** 7
**Confidence:** 4

**Review:**

The authors present hypotesis tests for testing the missingness process of the data (MCAR, MAR or MNAR) in predictive models. They use information theoretic approaches which have the advantage of handling heterogenous data types and of being non-parametric (in a specific sense). In particular, one of the proposed hypothesis tests allow to distinguish M(C)AR and MNAR mechanisms.

Testing the missingness mechanism is a key challenge and the proposed approach is novel, that is why I find that this paper has a strong interest for the workshop.

However, I think the differences with existing hypothesis tests are not explained enough. The proposed approach could have been compared with them in the experiments part. In addition, the authors could detail the computation times, when they say that running these tests is computationally expensive.

---

### Official Review · AnonReviewer1 · 2020-06-22
**Hypotheses tests for missingness processes**

**Confidence:** 3
**Rating:** 7

**Review:**

The authors propose two information-theoretic imputation tests for assessing the missingness process in partially observed data in prediction models. Importantly, using mutual information is advantageous as it is invariant to certain transformations and easily applicable to heterogeneous data.

It would be interesting to understand the limits of the proposed approach in more detail by evaluating the performance of the proposed tests on higher-dimensional data and to understand errors because of problems of the estimation of MI. Details for understanding the experiments in detail (e.g., NN size) are missing. A comparison with other methods would improve the paper.

---

### Decision · Program_Chairs · 2020-07-02

**Decision:**

Accept

**Comment:**

We're happy to accept this paper at Artemiss. We'll contact you soon to inform you about more details concerning the format of your presentation at the workshop, and the camera-ready version deadline. Please take into account the referee's comments to write the camera-ready version.